

# A genetically engineered mouse model for ovarian hyperstimulation syndrome

Wenkai Bi[1,2,3,*], Xinchen Jin[1,2,*], Shanshan Wu[1,2], Jing Wang[1,2], Xinhuan Su[4], Ling Gao[1,2] and Zhao He[1,2]

[1] Department of Endocrinology, Shandong Provincial Hospital & Medical Integration, and Practice Center, Shandong University, Jinan, Shandong, China
[2] Key Laboratory of Endocrine Glucose & Lipids Metabolism and Brain Aging, Ministry of Education, Shandong Key Laboratory of Endocrinology and Lipid Metabolism, Shandong Institute of Endocrine and Metabolic Diseases, Shandong Clinical Research Center of Dia, Jinan, Shandong, China
[3] Department of Nuclear Medicine, Shandong Provincial Hospital affiliated to Shandong First Medical University, Jinan, Shandong, China
[4] Department of Endocrinology, Department of Geriatrics, Shandong Provincial Hospital Affiliated to Shandong First Medical University, Jinan, Shandong, China
* These authors contributed equally to this work.

Corresponding author
Zhao He, zhaohe@sdu.edu.cn

## ABSTRACT

Ovarian hyperstimulation syndrome (OHSS) is a common iatrogenic complication resulting from ovarian stimulation in assisted reproductive technology (ART). Excessive stimulation by follicle-stimulating hormone (FSH) has been recognized as a primary cause of OHSS. However, understanding the pathophysiological mechanisms of OHSS and developing effective drugs have been limited due to the absence of suitable animal models. In this study, we generated a FSH gene knock-in (FKI) mouse model to inducible FSH expression by the Tet-on system, that reflects the clinical manifestations of OHSS in patients. Upon administration of doxycycline (Dox), the FKI mice exhibited significantly elevated serum FSH levels compared to their wild-type (WT) littermate controls, accompanied by increased levels of estradiol (E2) and luteinizing hormone (LH), ovarian enlargement, and enhanced peritoneal permeability. Thus, the FKI mouse model is a valuable tool for studying OHSS, particularly dissecting the pathophysiological mechanism and developing potential prevention strategies.

## INTRODUCTION

Ovarian hyperstimulation syndrome (OHSS) is a severe iatrogenic complication that often occurs during *in vitro* fertilization (IVF) and embryo transfer procedures (*Cao et al., 2022*). The development of OHSS involves a complex pathophysiological process. The clinical manifestations of OHSS include ovarian enlargement, electrolyte imbalances, gastrointestinal discomfort, ascites, pleural effusion, oliguria, thrombosis, and renal failure. The administration of ovulation-stimulating agents, such as follicle-stimulating hormone (FSH) (*Broekmans, 2019*), promotes the development of multiple ovarian follicles, which will result in ovarian enlargement and alterations in the secretion of various substances. In

addition to increased levels of estrogen and progesterone, there is also a significant elevation of vascular growth factors in the body. These elevated vascular growth factors stimulate vascular proliferation and increase capillary permeability (*Hulde et al., 2021*), leading to hemoconcentration and multi-system failure. Thus, assessment of the potential severity of OHSS, prevention and timely intervention for OHSS are critical during the ovulation induction process. However, the pathogenesis of OHSS remains incompletely understood due to the lack of suitable animal models. As a result, patients of OHSS are typically managed with supportive therapy and other approaches, which present limited therapeutic efficacy. The most common model for studying OHSS is exogenous gonadotropin-induced OHSS, which leads to variability in serum drug levels among individuals within the same group. Therefore, the development of a stable and reliable OHSS model is urgently for studying the pathogenesis of OHSS and developing treatment strategies.

Follicle-stimulating hormone (FSH) is an essential hormone in ovarian folliculogenesis. Even during the pre-antral (gonadotropin-independent) phase of follicle growth, FSH plays an active role in promoting follicular development. During the antral (gonadotropin-dependent) phase, FSH-dependent activation of aromatase induces the production of estradiol (E2), leading to the formation of large antral follicles (*Dewailly et al., 2016*). Exogenous FSH has been widely used as an ovarian stimulation agent to obtain a sufficient number of oocytes for *in vitro* fertilization. However, high doses of FSH are associated with an increased risk of OHSS (*Broekmans, 2019*). It raises the possibility that inducible FSH overexpression could be a valuable tool for studying OHSS. In this study, we utilized CRISPR-Cas9 technology to generate a transgenic mouse model with inducible FSH overexpression, which is triggered by doxycycline (Dox) administration. After 1 week of Dox supplementation, the mice exhibited typical OHSS manifestations, including ovarian enlargement and increased vascular permeability. Thus, the FKI mouse strain is a reliable, controllable, and time-save mouse model for studying the pathogenesis and developing potential preventive strategies for OHSS.

## MATERIALS AND METHODS

### Mice model and induction

*Cga* and *Fshb* are the two subunit-encoding genes of FSH. To generate mice with inducible FSH expression, the cDNAs of *Cga* and *Fshb* genes were respectively inserted on each side of a regulatory cassette containing pTRE (Tight BI), followed by the rtTA (reverse tetracycline-controlled transactivator) gene (Fig. S1A). The pTRE-Cga-Fshb-CAG-rtTA fragment was inserted at the H11 locus of C57BL/6J mice by using CRISPR/Cas9 technology (Fig. S1B). The mice were generated by the Nanjing Biomedical Research Institute at Nanjing University (Nanjing, China).

The FKI mice were bred with wild-type (WT) C57BL/6J mice to maintain the strain (Fig. S1C). After genotyping (Fig. S2 and Tables S1–S4), female FKI mice and WT littermate controls were exposed to 1 g/L Dox supplementation (Sinopharm, Shanghai, China) in their drinking water starting at 8 weeks of age. Besides Dox induction, no other experimental interventions were carried out, and no analgesia was given. After 1 week of

Dox stimulation, vascular permeability was measured, the ovary and serum samples of FKI and WT mice were obtained.

Animal anesthesia was performed *via* intraperitoneal injection of 1% sodium pentobarbital (0.01 mL/10 g body weight) prior to blood collection. The retro-orbital venous plexus blood sampling procedure was conducted as follows: Subjects were positioned laterally with cephalic fixation using a stereotaxic apparatus. The target area at the mandibular venous plexus was disinfected with 75% ethanol. A sterile 23-gauge needle was vertically inserted into the hairless subcutaneous depression while maintaining mandibular palpation to prevent osseous perforation. Cephalic positioning below cardiac level was maintained to optimize venous pressure. Whole blood samples were collected using heparinized capillary tubes immediately upon needle withdrawal. Hemostasis was achieved through 2-min digital compression with sterile gauze.

The ovaries were obtained as follows: A skin incision approximately 0.5–1 cm in length was made along the midline of the mouse's back with surgical scissors. The psoas muscle was cut 1 cm below the ribs of the spine, and the adipose tissue surrounding the ovaries were visualized. Then, the ovaries were removed by dragging them out of the incision and clamping the fat tissue.

There were about 10 mice per group in each analysis. The number of mice was justified by pre-experiment. The mice were housed at 24 °C on a 12-h:12-h light/dark cycle and provided standard chow and water *ad libitum*. Animal welfare is ensured during the process, and the animals are properly disposed of at the end of the process.

The animal experiments were performed in accordance with the guidelines and approval of the Animal Ethics Committee of Shandong Provincial Hospital (NSFC: NO. 2021-274).

## Histology and immunohistochemistry

Mouse ovaries were fixed in 4% paraformaldehyde (DF0135; Leagene Biotechnology, Nanjing, China) at room temperature for 24 h, embedded in paraffin, and sectioned at a thickness of 5 μm. The sections were then mounted on glass slides, and ovarian tissue morphology was examined by using hematoxylin and eosin (H&E) staining. In the follicle diameter measurement, the follicles of the mature follicle stage were measured. The average diameter of each mature follicle per mouse was calculated and analyzed.

## Determination of vascular permeability

Vascular permeability was assessed by using a modified Miles assay (*Brash, Ruhrberg & Fantin, 2018*). Briefly, 0.1 mL of 5 mM Evans blue dye was injected intravenously into the mice. After 30 min, the mice were anesthetized, and 2 mL of saline was injected intraperitoneally. The mouse abdomen was gently massaged for 1 min to mix the dye, after which 1 mL of ascites was collected and centrifuged at 12,000 rpm. Then, 100 μl of the supernatant was transferred to a 96-well ELISA plate for absorbance measurements at 405 nm (for calibration) and 620 nm. Following perfusion with 20 mL of saline to remove blood, the ovaries were harvested and incubated in 2 mL of formamide for 24 h at 65 °C. Absorbance of the formamide was measured by using a spectrophotometer.

## ELISA

VEGF, IL-6, and IL-11 levels in serum or ovarian tissue lysates were determined using ELISA kits (EM009, EM004, EM012; ExCell Biotech, Suzhou, China), following the manufacturer's protocols. Follicle-stimulating hormone (FSH), estradiol (E2), luteinizing hormone (LH), testosterone (T), thyroxine (T4), thyroid-stimulating hormone (TSH), and glucocorticoid (GC) levels were measured by using ELISA kits (CSB-E06871m, CSB-E05109m, CSB-E12770m, CSB-E05101m, CSB-E05083m, CSB-E05116m; CUSABIO, Wuhan, China) and SBJ-M0964 (SenBeiJia, Wuhan, China).

## Metabolism cage

Energy expenditure and locomotor activity were recorded by using the Oxymax system (Columbus Instruments, Comprehensive Lab Animal Monitoring System (CLAMS)). Mice were housed individually under a 12-h light/12-h dark cycle at an ambient temperature of 24 ± 2 °C.

## Statistical analysis

Data were analyzed using GraphPad Prism 7 (GraphPad Software, Boston, MA, USA). Results are presented as mean ± SD. Statistical significance was determined using an unpaired two-tailed Student's t-test or one-way ANOVA for comparisons across multiple groups. Kaplan–Meier survival analysis was performed using the log-rank test. A $p$-value < 0.05 was considered statistically significant.

# RESULTS

## Higher serum FSH level is associated with ovary enlargement

Female FKI and WT mice were exposed to Dox for 1 week. No difference in serum FSH levels was observed in WT mice before and after induction. In contrast, FKI mice with Dox supplementation showed a significant increase in serum FSH levels, indicating conditional overexpression of FSH in the FKI mouse model (Fig. 1A). After induction, FKI mice also exhibited ovarian enlargement, as revealed by increased ovarian weight (Figs. 1B, 1C) and larger follicle diameters compared to WT mice (Figs. 1D–1F). Moreover, some ovaries of FKI mice were obviously hemorrhagic (Fig. 1B), consistent with large congested follicles (Fig. 1E). These phenotypes in the FKI mouse model are typical manifestations of OHSS, suggesting that FKI mice is an effective animal model for OHSS research.

## FSH overexpression elicits abnormal hormone levels and altered blood cell profile

Next, we examined the hormone profile in FKI mice with overexpressing FSH. We found that estrogen (E2), luteinizing hormone (LH), and testosterone (T) levels were significantly higher in FKI mice compared to WT mice (Figs. 2A–2C), which is consistent with other OHSS models (Castillo et al., 2020; Kasum, 2010; van de Lagemaat et al., 2011). The T4 and TSH levels were measured in FKI mice to explore the effect of OHSS on thyroid function. FKI mice exhibited increased thyroxine (T4) levels, and unchanged thyroid-stimulating

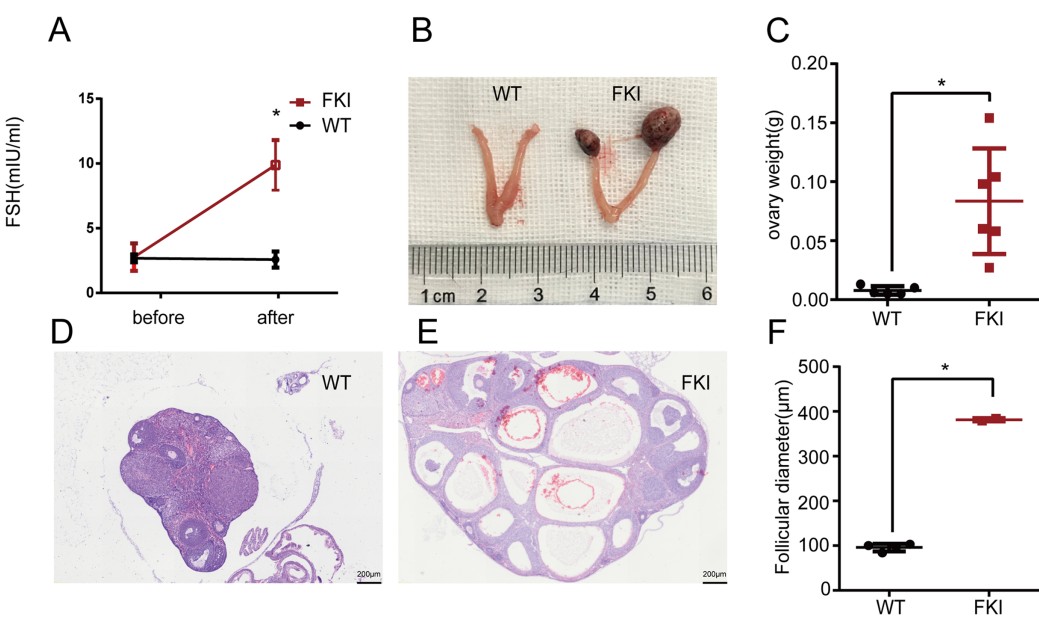

**Figure 1 Female FKI mice showed OHSS manifestation.** (A) Serum FSH levels in WT and FKI mice before and after 1 week of Dox supplement. $n$ = 9/WT group, $n$ = 10/FKI group. (B) Representative images of ovaries of WT mice and FKI mice after Dox induction. (C) Weight of ovaries. (D) H&E staining of ovary of WT mice. Scale bars = 200 µm. (E) H&E staining of ovary of FKI mice. Scale bars = 200 µm. (F) Quantitative analysis of follicle diameter. Data are presented as the mean ± SD. *$P < 0.05$.

hormone (TSH) (Figs. 2D, 2E), implying that thyroid function might adjust to the state of OHSS.

Systemic inflammation has been recognized as a cause of OHSS. Routine blood analysis showed no differences in red blood cell (RBC) count between WT and FKI mice (Fig. 2F), but platelet (PLT) and white blood cell (WBC) counts were lower in FKI mice (Figs. 2G, 2H). The decrease in PLT may be linked to hypercoagulability and abnormal hormone levels in OHSS. Lymphocyte (LYM) levels, subspecies of WBC, were significantly lower in FKI mice than in WT mice (Fig. 2I). Although no difference in neutrophil (NEU) was observed between the FKI and WT mice (Fig. 2J), the neutrophil-lymphocyte ratio (NLR) and neutrophil-platelet ratio (NPR), the indicators of inflammation, were higher in FKI mice than that in WT mice (Figs. 2K, 2L).

## FKI mice with elevated FSH levels show increased vascular permeability

Increased vascular permeability is a hallmark of OHSS, potentially leading to hemoconcentration and life-threatening conditions. After 1 week of Dox supplementation, FKI mice exhibited increased capillary permeability in the peritoneum and ovaries (Figs. 3A, 3B), with unchanged plasma osmotic pressure and hematocrit (HCT) (Figs. 3C, 3D). Vascular permeability in OHSS is primarily caused by inflammatory factors such as VEGF and IL-6, however, no differences in VEGF, IL-6, IL-11, and glucocorticoid (GC) levels

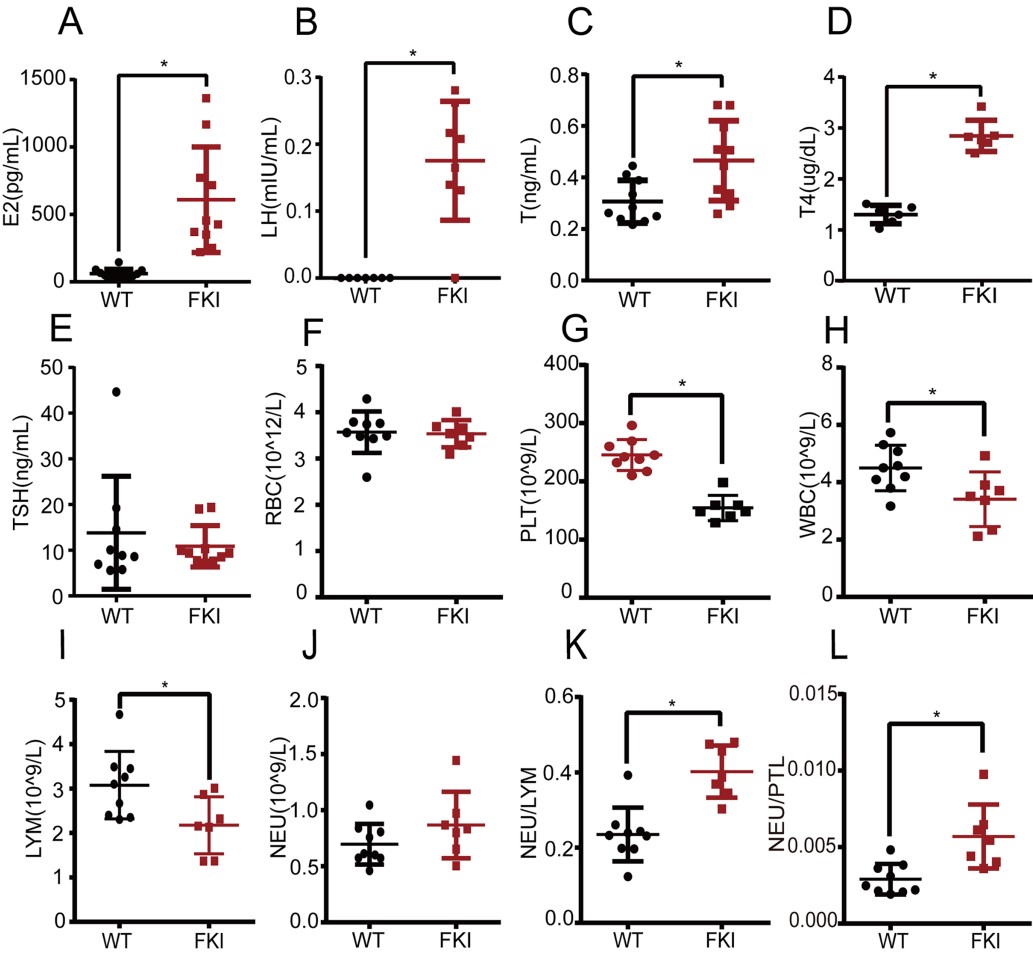

**Figure 2 Female FKI mice exhibited disturbed hormone levels and altered levels of blood cells.**
(A) Serum estradiol (E2) level. (B) Serum luteinizing hormone (LH) level. (C) Serum testosterone (T) level. (D) Serum thyroxine (T4) level. (E) Serum thyroid-stimulating-hormone (TSH) level. (F) The number of red blood cells (RBC). (G) The number of platelets (PLT). (H) The number of white blood cells (WBC). (I) The number of lymphocytes (LYM). (J) The number of neutrophils (NEU). (K) Neutrophil-Lymphocyte ratio (NLR). (L) Neutrophil-Platelet ratio (NPR). Data are presented as the mean ± SD. *$P < 0.05$.

were observed between WT and FKI mice (Figs. 3E–3H), suggesting that other factors are responsible for the increased permeability in FKI mice.

## FKI mice with Dox supplement exhibited reduced activity and heat production

To further characterize the mice model, we assessed the metabolism of FKI mice. During both nighttime and daytime phases, FKI mice displayed reduced physical activity and heat production compared to WT controls (Figs. 4A–4D), indicating a hypometabolic state. However, no differences in body weight and food intake were observed between the two groups (Figs. 4E, 4F), indicating the global hypoactivity and hypometabolism in FKI mice

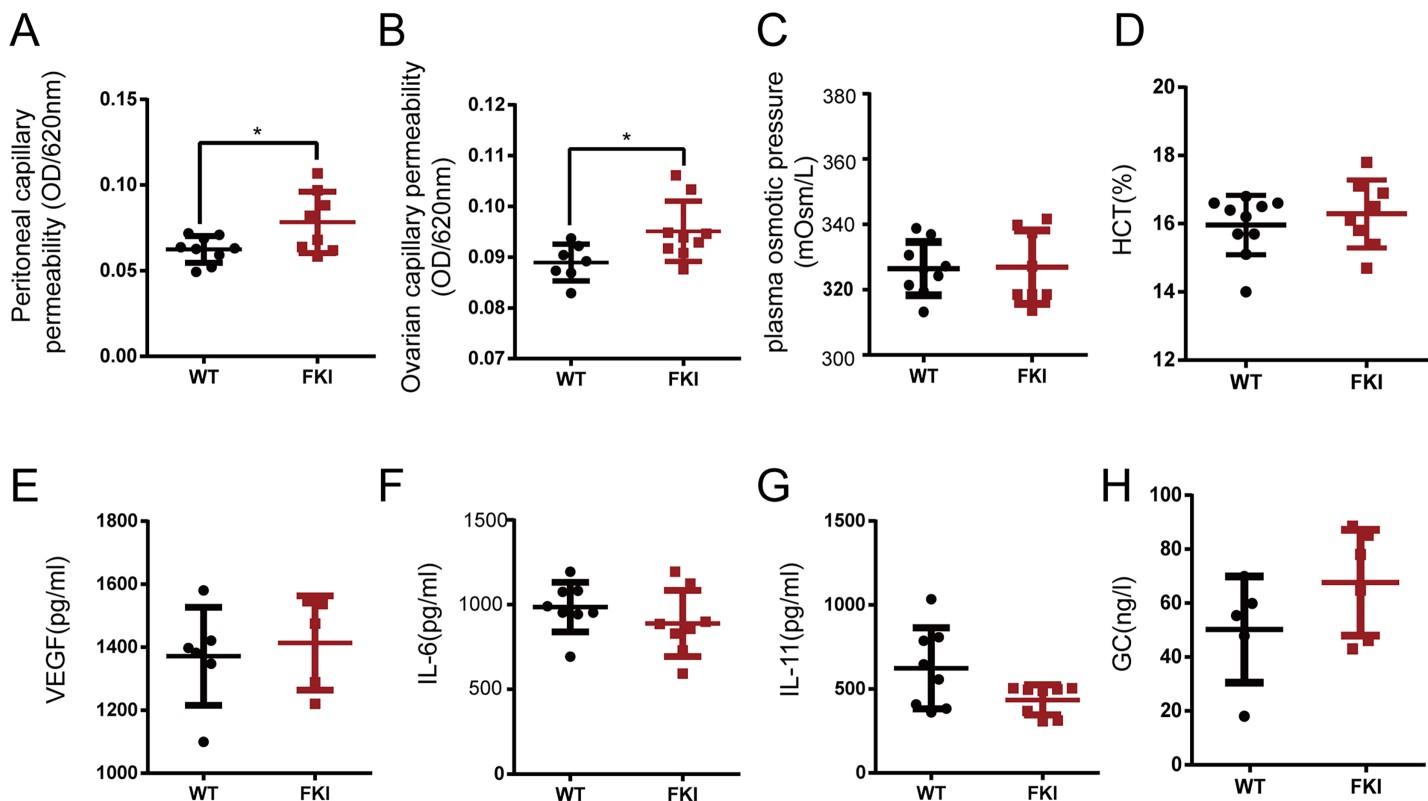

**Figure 3  The FKI female mice increased vascular permeability.** (A) The normalized absorbance values (optical density, OD) of ascites. (B) The normalized absorbance values (optical density, OD) of Evans blue extracted in formamide from ovarian tissue. (C) Plasma osmotic pressure. (D) Blood hematocrit (HCT). (E) Serum VEGF level. (F) Serum IL-6 level. (G) Serum IL-11 level. (H) Serum glucocorticoid (GC) level. Data are presented as the mean ± SD. *$P < 0.05$.               

were caused by systemic inflammation, which is similar to the manifestation of OHSS patients.

## FKI mice with elevated FSH levels exhibit weight loss and reduced lifespan

Subsequently, the long-term effects of OHSS on FKI mice were investigated. Eight-week-old FKI mice were exposed to Dox in drinking water to induce FSH overexpression. After 6 weeks of induction, FKI mice exhibited significantly higher FSH levels and lower body weight than WT mice (Figs. 5A–5C). Notably, after 10 weeks of induction, FKI mice showed a reduced survival rate (Fig. 5D), implying that prolonged elevated FSH levels may lead to mortality or late-stage OHSS.

## DISCUSSION

To investigate the underlying mechanism of OHSS and develop prevention strategies for OHSS, various animal models have been established by repeated injections of exogenous ovarian-stimulating drugs. For example, a rabbit model of OHSS was developed by continuous high-dose injection of human menopausal gonadotropin (HMG), which takes a long experimental time and high costs (*Polishuk & Schenker, 1969*). Another model of OHSS

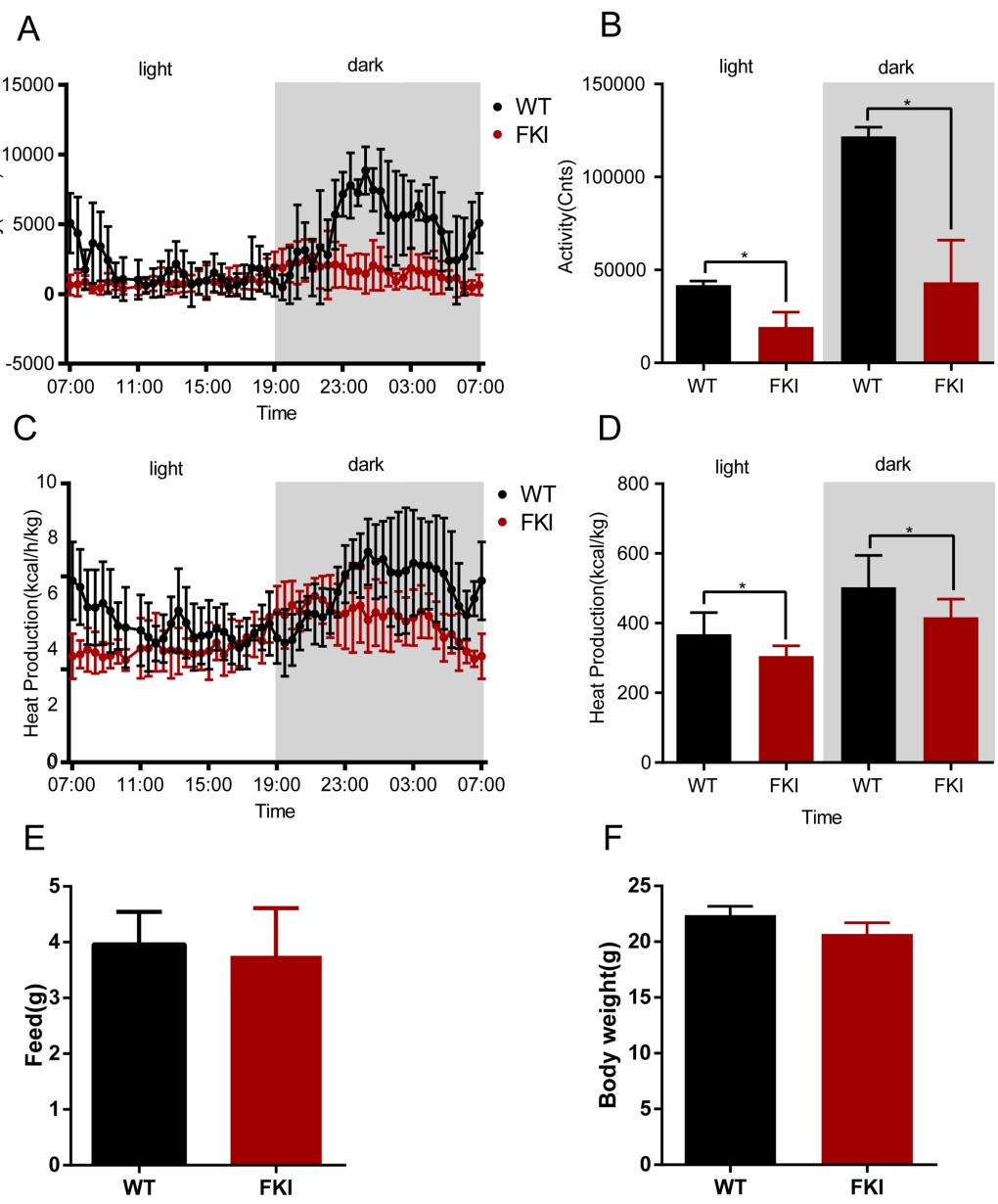

**Figure 4 The activity and metabolism of FKI mice were decreased after Dox induction.** (A) Activity of mice during the day phase (12 h) and dark phase (12 h). (B) Area under curve of activity during the day phase (12 h) and dark phase (12 h). (C) Whole-body thermogenesis of mice during the day phase (12 h) and dark phase (12 h). (D) Area under curve of whole-body thermogenesis during the day phase (12 h) and dark phase (12 h). (E) Food intake of mice in 24 h. (F) Body weight of mice after Dox induction. Data are presented as the mean ± SD. *$P < 0.05$. $n$ = 8/WT group, $n$ = 12/FKI group.

is generated by combining HMG with HCG (*Oriowo, 2004*), which is more similar to the clinical manifestation. Similarly, equine chorionic gonadotropin (ECG) and HCG were used to establish a rat model of OHSS *via* different injection approaches (*Ujioka et al., 1997*), and intraperitoneal injections of multiple hormones (PMSG, HMG, FSH, and HCG) were used

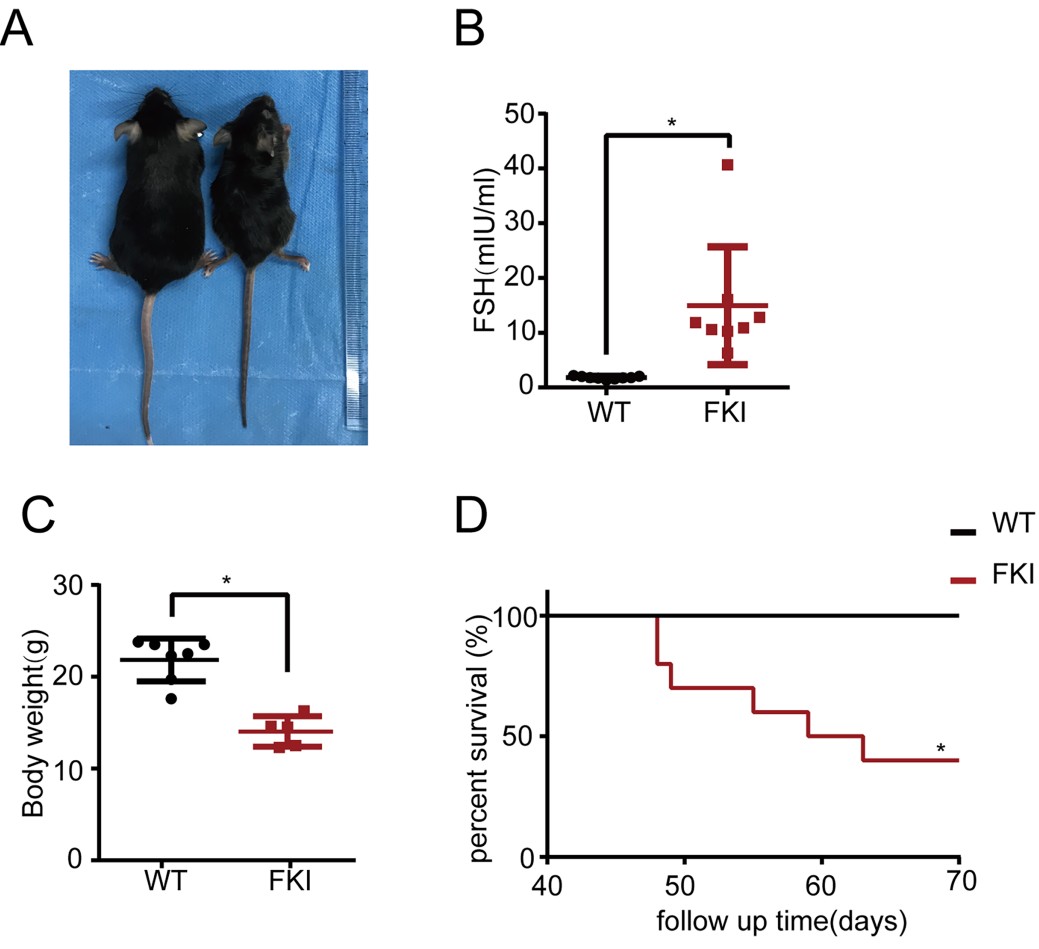

**Figure 5 The FKI female mice with long-term of Dox induction exhibited weight loss and reduced lifespan.** (A) Representative images of female WT and FKI mice with 6 weeks of Dox induction. (B) Serum FSH levels of WT and FKI female mice with 6 weeks of Dox induction. Data are presented as the mean ± SD. *$P < 0.05$. (C) Body weight of WT and FKI female mice 6 weeks of Dox induction. Data are presented as the mean ± SD. *$P < 0.05$. (D) Kaplan–Meier survival curves of female WT and FKI mice with long term of Dox induction. $n = 10$/per group. *$P < 0.05$.

to develop another rat model (*Ajonuma et al., 2005*). A rhesus monkey model was also created by using exogenous hormone injections (*Molskness et al., 2004*). However, these models are required to take more time and money cost, and have variability in drug levels within individuals.

In our FKI mouse model, two FSH subunit-encoding genes, *Cga* and *Fshb*, were inserted in the control of a regulatory cassette, allowing for conditional FSH overexpression upon Dox supplementation. Before Dox induction, FSH levels were similar between FKI and WT mice. After 1 week of Dox induction, FKI mice exhibited increased FSH levels, ovarian enlargement (Fig. 1), and enhanced vascular permeability (Fig. 3), which are consistent with OHSS clinical features. Importantly, FKI mice can be bred with wild-type mice to obtain experimental animals, providing a short reproductive cycle and cost, thereby saving

time and improving experimental reliability. Thus, the FKI model is a suitable and cost-effective tool for studying the pathophysiology and treatment of OHSS. Moreover, the inducible mouse model could be used to induce varying responses through different dose and duration of Dox induction. The varying degrees of severity in OHSS might be underpinned by distinct molecular mechanisms. The FKI model would provide a new approach and tool for the further researches. In addition, literature showed that FSH play a significant role in extragonadal physiological functions, including promoting hepatic gluconeogenesis, adipocytic lipid biosynthesis, postmenopausal osteoporosis and mood regulation (*Bi et al., 2020*; *Gera et al., 2022*; *Guo et al., 2019*). This FSH gene knock-in mouse model might be employed to explore the influence of FSH on other systems in subsequent researches.

In patients with OHSS, the changes in levels of LH and FSH were complex, being influenced by multiple factors, including the protocol, duration and dosage of drug administration. Due to the abnormal ovarian function and the disorder of the hormonal feedback regulation mechanism, the levels of LH and FSH usually lost the typical variation pattern of the ovulatory cycle. Similarly, the production of androgens was often upregulated in response to ovarian hyperstimulation. Nevertheless, the effect of androgens might be not apparent due to significant elevation of estrogen levels. After Dox stimulation, the FKI mice showed elevated levels of FSH, LH, E2, and T, which is similar with those of patients with OHSS. The underlying mechanism of these changed hormone levels warranted further studies. The increased LH levels in FKI mice was probably due to the increasing subunit availability. As known, FSH and LH possess an identical α subunit (Cga), which is encoded by the same gene and is responsible for the basic framework for binding to the receptor. We hypothesized that the increased expression of *Cga* gene in FKI mice might be responsible for the higher level of LH. The increased T levels of FKI mice might due to the response of interstitial cells to higher LH levels.

Compared with WT mice, the FKI mice exhibited a disorder in thyroid function. Although some cases of thyroid dysfunction had been reported in OHSS patients (*Bachmakova et al., 2014*; *Kasum, 2010*; *Poppe et al., 2011*; *Skweres et al., 2014*), there is still no consensus on thyroid function in OHSS. Literature shows no additional effects of OHSS on thyroid function (*Poppe et al., 2011*), in contrast, other literature suggests a clinical and molecular association between thyroid hormones and gonadotropins, including FSH and HCG (*Casarini et al., 2016*). Consistently, there is an interaction between OHSS and thyroid dysfunction, as case reports show patients with both hypothyroidism and OHSS (*Zhou et al., 2024*). Of note, TSH and FSH receptors share the same α-subunit, thereby elevated thyroid peroxidase antibodies (TPOAb) could contribute to OHSS development (*Kilpatrick et al., 2014*). Thus, the FKI model provides a valuable tool for investigating the link between OHSS and thyroid function.

Inflammation plays a critical role in OHSS pathogenesis. Various hematological markers have been used to assess systemic inflammation in OHSS (*Tamhane et al., 2008*). NLR reflects immune and inflammatory status, as neutrophils is an indicator of non-specific inflammation and lymphocytes play a defensive and regulatory role in inflammatory response. Moreover, NLR has been identified as an independent predictor of

OHSS development (*Baser et al., 2022*; *Verit et al., 2014*). In this study, FKI mice exhibited elevated NLR, which is similar to OHSS patients. Elevated NLR is also associated with various diseases, such as myocardial infarction (*Kahraman et al., 2021*; *Tudurachi et al., 2023*), metabolic syndrome (*Marra et al., 2023*) and hypertension (*Meng et al., 2024*). Notably, OHSS is also reported to be related to these diseases. Myocardial infarction is a secondary outcome of OHSS due to platelet activation and thrombosis (*Akdemir, Uyan & Emiroglu, 2002*; *Somigliana et al., 2014*). Lipid metabolic disorders are associated with the development of OHSS (*Liu et al., 2020*). Hypertension in pregnancies is usually associated with OHSS (*Dobrosavljevic & Rakic, 2020*). The FKI model is, thus, a useful tool for studying the inflammatory components of OHSS and related disorders. In addition, NPR is an important indicator of thrombosis and inflammation severity. The interaction between neutrophils and platelets involves various mechanisms, including neutrophil-platelet aggregates formation, and platelet activation (*Zhang & Lang, 2024*). Overall, the increased NLR and NPR in FKI mice suggest a severe inflammatory response and hypercoagulability, which are consistent with OHSS manifestations.

In the present study, both short-term and long-term effects of OHSS were investigated. Interestingly, FKI mice subjected to short-term Dox supplementation (1 week) exhibited a hypoactive state (Fig. 4), whereas long-term exposure (6 weeks) resulted in significant lifespan shortening (Fig. 5). These data suggested that the model might provide a powerful tool for exploring a spectrum of short/long-term physiological and pathological consequences, including metabolic dysregulation, organ function impairment, and behavioral alterations, thus advancing our comprehensive understanding of OHSS.

## CONCLUSION

In this study, a mouse model with conditional FSH overexpression (FKI mice) was generated. After 1 week of Dox induction, FKI mice displayed clinical and pathological features of OHSS, including ovarian enlargement, elevated FSH and E2 levels, and increased vascular permeability. These findings indicate that FKI mice is a valuable model in studying the pathogenesis of OHSS and developing potential preventive treatments for OHSS.

### Funding

This research was supported by National Science and Technology Major Project (2023ZD0506800) and National Natural Science Foundation of China (NSFC), Grant Number: 82270899. This work was supported by the Shandong Provincial Natural Science Foundation (ZR2022MH121 and ZR2021QH150). The funders had no role in study design, data collection and analysis, decision to publish, or preparation of the manuscript.

### Grant Disclosures

The following grant information was disclosed by the authors:
National Science and Technology Major Project: 2023ZD0506800.

National Natural Science Foundation of China (NSFC): 82270899.
Shandong Provincial Natural Science Foundation: ZR2022MH121, ZR2021QH150.

## Competing Interests

The authors declare that they have no competing interests.

## Author Contributions

- Wenkai Bi conceived and designed the experiments, authored or reviewed drafts of the article, and approved the final draft.
- Xinchen Jin performed the experiments, authored or reviewed drafts of the article, and approved the final draft.
- Shanshan Wu performed the experiments, authored or reviewed drafts of the article, and approved the final draft.
- Jing Wang analyzed the data, prepared figures and/or tables, authored or reviewed drafts of the article, and approved the final draft.
- Xinhuan Su analyzed the data, prepared figures and/or tables, and approved the final draft.
- Ling Gao conceived and designed the experiments, authored or reviewed drafts of the article, and approved the final draft.
- Zhao He conceived and designed the experiments, authored or reviewed drafts of the article, and approved the final draft.

## Animal Ethics

The following information was supplied relating to ethical approvals (*i.e.*, approving body and any reference numbers):

The animal experiments were performed in accordance with the guidelines and approval of the Animal Ethics Committee of Shandong Provincial Hospital.

## Data Availability

The raw data is available in the Supplemental Files.

## Supplemental Information

Supplemental information for this article can be found online at http://dx.doi.org/10.7717/peerj.19531#supplemental-information.

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
