# Peer review of "A genetically engineered mouse model for ovarian hyperstimulation syndrome"

_PeerJ, doi:10.7717/peerj.19531_

## Round 0.1 · original submission · Major Revisions

Your article has been enthusiastically received by the reviewers, who although they have many comments and issues for you to address, clearly recognise the merit in your study and are keen for it to be published, as indeed am I.

Please address carefully all comments they make and modify your paper clearly and succinctly in line with these. The majority of the points raised are with the writing and the text needs significant work.

The suggestion that you add control/wildtype histology and moving Suppl figure 3 to the main manuscript should be prioritized.

I look forward to the revision.

**Language Note:** The review process has identified that the English language must be improved. PeerJ can provide language editing services - please contact us at [email protected] for pricing (be sure to provide your manuscript number and title). Alternatively, you should make your own arrangements to improve the language quality and provide details in your response letter. – PeerJ Staff

Reviewer 1 ·

Basic reporting

-

Experimental design

Response to stimulation can vary, while the method of measuring the diameter of follicles and serum hormone levels is not specified during follow-up after stimulation.

Validity of the findings

Response to stimulation displayed features of OHSS. Baseline hormonal levels are not documented.

Additional comments

FKI mice are a valuable model for studying the pathogenesis of OHSS and developing potential preventive treatments, as they exhibit clinical and pathological features such as ovarian enlargement and elevated FSH and E2 levels. However, the response to stimulation may vary, influencing the severity of OHSS. Further research utilizing models with varying responses to stimulation may be useful in assessing OHSS severity and improving its management.

A few corrections are highlighted in Pink color for corrections.

Annotated reviews are not available for download in order to protect the identity of reviewers who chose to remain anonymous.

Reviewer 2 ·

Basic reporting

-The writing needs improvement. Some parts read smoothly, but others need editing for English grammar.

-The Methods starting line 91 are written like a recipe or protocol and need to be rewritten in past tense in a format that is expected for a manuscript.

-Lines 107-114 are details that are not needed and can be deleted.

-Figures are good. Raw data is shared in supplemental files.

-In the Results, there is too much discussion which should be moved to later in the Discussion section.

-The Discussion is insufficient. There is a large amount of data that has not been discussed. For example, yes, E2, LH and T were increased in FKI mice, but why? Your model is different from those previously reported so you should explain. FSH would increase E2, but does E2 increase LH or is it through increasing subunit availability to make LH, or both - interpret based on the previous literature? If LH is up, then it may be responsible for the OHSS as clinically hCG treatment is most often the cause of increased VEGF and OHSS.

Experimental design

Although gonadotropin overexpressing mice have been previously made and evaluated, to my knowledge this study is the first to do this with an inducible system. Such systems are important as one can bypass the effects of overexpression during early development.

Controls were appropriately treated with DOX.

Number of mice appears adequate (n=11 to 12).

Line 156 - Results Figure 1 section 3.1 - there should be more detail written about the anatomy & histology. The ovaries are obviously hemorrhagic and large follicles look like they are blood filled, was this so?

Validity of the findings

Show a control ovary/oviduct from wildtypes for comparison, like it done for Fig 1 d and e FKI mice.

Rigor is adequate. Findings are valid.

Very few conclusions are made other than this would be a good rodent model for OHSS.

Additional comments

-I like this manuscript a lot. This manuscript has some great data but the way the text is written the data is underwhelming.

-line 29 “inducible” should be “induced” the way this sentence is written
-line 76-77 You refer to the alpha and beta subunit genes. Is this the whole gene or rather the cDNA?
-In figure legends special character < did not appear in the PDF
-I would like to see Supplemental Figure 3 in the main body of the manuscript Results and Discussed. The reduced weight, activity and heat production in FKI mice suggest T3 might be down but T4 was not different.

-Lines 171-173 would be better in the Discussion

---

## Round 0.2 · accepted · Accept

Thank you for addressing the issues raised. I am satisfied with these changes and/or rebuttal comments and so am now happy to accept this paper.